# The Use of Lanthanum Ions and Chitosan for Boron Elimination from Aqueous Solutions

**DOI:** 10.3390/polym11040718

**Published:** 2019-04-19

**Authors:** Joanna Kluczka, Gabriela Dudek, Alicja Kazek-Kęsik, Małgorzata Gnus, Maciej Krzywiecki, Krzysztof Mitko, Katarzyna Krukiewicz

**Affiliations:** 1Department of Inorganic, Analytical Chemistry and Electrochemistry, Faculty of Chemistry, Silesian University of Technology, B. Krzywoustego 6, 44-100 Gliwice, Poland; alicja.kazek-kesik@polsl.pl (A.K.-K.); krzysztof.mitko@polsl.pl (K.M.); 2Department of Physical Chemistry and Technology of Polymers, Faculty of Chemistry, Silesian University of Technology, ks. M. Strzody 9, 44-100 Gliwice, Poland; gabriela.maria.dudek@polsl.pl (G.D.); malgorzata.gnus@polsl.pl (M.G.); katarzyna.krukiewicz@polsl.pl (K.K.); 3Institute of Physics—Centre for Science and Education, Silesian University of Technology, Konarskiego 22B, 44-100 Gliwice, Poland; maciej.krzywiecki@polsl.pl

**Keywords:** chitosan hydrogel beads, bio-adsorbent, lanthanum hydroxide, boron removal

## Abstract

Boron is an essential element for plants and living organisms; however, it can be harmful if its concentration in the environment is too high. In this paper, lanthanum(III) ions were introduced to the structure of chitosan via an encapsulation technique and the obtained hydrogel (La-CTS) was used for the elimination of the excess of B(III) from modelling solutions. The reaction between boric acid and hydroxyl groups bound to the lanthanum coordinated by chitosan active centres was the preponderant mechanism of the bio-adsorption removal process. The results demonstrated that La-CTS removed boric acid from the aqueous solution more efficiently than either lanthanum hydroxide or native chitosan hydrogel, respectively. When the initial boron concentration was 100 mg/dm^3^, the maximum adsorption capacity of 11.1 ± 0.3 mg/g was achieved at pH 5 and the adsorption time of 24 h. The successful introduction of La(III) ions to the chitosan backbone was confirmed by Scanning Electron Microscopy with Energy Dispersive X-Ray Spectroscopy, Fourier-Transform Infrared Spectroscopy, X-Ray Diffraction, X-ray Photoelectron Spectroscopy, and Inductively Coupled Plasma Optical Emission Spectroscopy. Due to its high-performance boron adsorption-desorption cycle and convenient form, La-CTS seems to be a promising bio-adsorbent for water treatment.

## 1. Introduction

Although boron is essential for plants and living organisms, its excess can be severely harmful to the environment. It is known that the main sources for boron contamination of water are wastewater discharges from the glass and ceramic industry (the largest market representing 56% global borate demand), but substantial amounts of this element are also emitted by metallurgy, mining and treatment of borate minerals, coal mining, and power industry. Boron is discarded to the environment as domestic wastewater effluents that may be extremely enriched in this element, with B(III) concentrations varying from several micrograms to several hundred micrograms per liter [1]. Contaminating drinking water, boron significantly affects the metabolism of Ca, P, and F, and can damage the nervous system in humans. The excess of boron can also lead to lowering or even destruction of crops [2]. This is why the recommended level of boron in drinking water according to the World Health Organization guidelines should not exceed 2.4 mg/dm^3^ [3]. In Poland, the limit is more restrictive and equals to 1 mg/dm^3^, both in natural water and in wastewaters discarded in the environment [4].

There are numerous methods proposed for the removal of B(III), including chemical precipitation, ion exchange, adsorption, electrocoagulation, solvent extraction, and membrane technologies, e.g., reverse osmosis, forward osmosis, nanofiltration, ultrafiltration, and membrane distillation [5,6,7,8,9,10]. Adsorption is considered to be a suitable choice, especially for treating wastewaters containing low concentration of boron, mainly because it is a rapid and economical method. New approaches to the water treatment by adsorption include the use of various bio-adsorbents, e.g., cellulose, pectin, alginate, and chitosan [11,12,13,14].

Chitosan (CTS) is a bio-renewable polymer derived from the alkaline deacetylation of chitin. It has excellent properties, including biocompatibility, biodegradability, non-toxicity, as well as bacteriostatic and fungistatic qualities [15]. CTS due to the unique properties has gained a place in many applications. Chitosan-based hydrogels as smart biomaterials are used in biomedical applications, such as drug delivery, tissue engineering, wound dressing, and biosensors [16,17,18,19]. In recent decades, chitosan-based hydrogels have been the object of many studies in other applications, among them in the field of environmental science and technology [20]. Due to its adsorption and ion exchange properties, CTS is studied for removal of heavy metal ions, dyes, pesticides, and pharmaceuticals [14,21]. As a result of its functionalization with N-, S-, and O-containing groups, CTS can also be used in the trace analysis to the selective recovery of different metals and metalloids from the salt matrix, e.g., natural waters, waste streams, and geological samples [22].

The presence of hydroxyl groups into the chitosan structure is reported to show good affinity towards borate, indicating the potential application of CTS for boron removal [23,24,25]. One way to introduce additional hydroxyl groups to the CTS backbone is to form a hybrid with a metal hydroxide that possesses a high boron affinity, which can be realized through the encapsulation technique. This technique for the preparation of hydroxide-CTS hybrids has been already developed and optimized to ensure low costs and high efficiency [26,27]. The encapsulation technique consists of the formation of metal hydroxide as regular beads in situ during the process of coagulation of CTS and metal salt solution in contact with NaOH solution. Many hydroxides and oxides are known from their potency to bind oxyborate, namely Ni(OH)_2_, Zn(OH)_2_, Co(OH)_2_, Mg(OH)_2_, Fe(OH)_3_, Al(OH)_3_, ZrO_2_, and TiO_2_ [27,28,29,30,31,32,33]. However, to date no lanthanum compounds have been used in this process.

Lanthanum is the third element of Group 3 and belongs to the rare earth elements—its content in the Earth’s crust equals 3.2 × 10^–3^ wt%. Although lanthanum has no biological role for human beings, it is also not particularly toxic. Being essential for some microorganisms, it has been found to exhibit antimicrobial activity and antitumor properties [34,35]. Lanthanum is used to produce Phoslock, the commercial bentonite clay in which the sodium or calcium ions are exchanged for La^3+^. The main application of Phoslock is the reduction of phosphorus, one of the major contributing factors to algal growth in lakes, ponds, and drinking water storage reservoirs [36,37]. Recently, lanthanum has been used for the functionalization of CTS to obtain sorbents for the removal of specific pollutants from water and sewage. Particularly, lanthanum and aluminium-lanthanum binary oxyhydroxides on CTS template were synthesized by the greener in situ one-pot method for the separation of fluoride from drinking water [38,39,40]. Preethi and Meenakshi [41] used chromate and fluoride for the design of single system La^3+^ impregnated chitosan/β-cyclodextrin biopolymeric materials. Microporous lanthanum-chitosan magnetic spheres were also successfully used for phosphate removal from water [42]. Other researchers used the La^3+^ entrapped chitosan bio-polymeric matrix for the recovery of oil from an oil-in-water emulsion [43]. In the view of the above reports, it seems highly probably that chitosan modified by lanthanum ions should enhance the adsorption of oxyborates, even though, according to our best knowledge, there are no literature reports describing the separation of oxyborate from aqueous solution using lanthanum(III).

Accordingly, the objectives of this paper are: (1) to synthesise chitosan attached with lanthanum(III) (La-CTS) by the method employing co-precipitation and coagulation; (2) to study the adsorption of boron on lanthanum hydroxide and on La-CTS hydrogel beads in a batch system; (3) to characterise the chemical and phase composition of La-CTS before and after adsorption using a Fourier-transform infrared spectrometry (FTIR), scanning electron microscopy (SEM), X-ray diffractometry (XRD), X-ray photoelectron spectroscopy (XPS), and inductively coupled plasma optical emission spectroscopy (ICP OES); (4) to consider the possible mechanism of the bio-adsorption of B(III) on La-CTS hydrogel beads.

## 2. Materials and Methods

### 2.1. Reagents

A basic standard solution of boron in the form of boric acid (1 g/dm^3^) and a basic standard solution of lanthanum in the form of lanthanum nitrate (1 g/dm^3^) were supplied by Merck KGaA, Darmstadt, Germany. Chitosan (molecular weight of 600,000–800,000 g/mol) was purchased from Acros Organics, Geel, Belgium; the degree of deacetylation of CTS, as determined by means of a H^1^ NMR method, was equal to 97%. Lanthanum nitrate, La(NO_3_)_3_·6H_2_O, was supplied by Reachim Ltd., Moscow, Russian Federation. Boric acid, sodium hydroxide, hydrochloric acid, and glacial acetic acid were purchased from Avantor Performance Materials Poland S.A., Gliwice, Poland. All the reagents employed in the study were analytical reagent grade.

### 2.2. Analysis and Apparatus

The concentrations of boron and lanthanum were determined using ICP-OES (inductively coupled plasma optical emission spectroscopy) with a Varian 710-ES spectrometer (Varian, Mulgrave, Victoria, Australia). The following emission lines were used: λ = 208.956, λ = 249.678 and λ = 249.772 nm (for boron) and λ = 379.082, λ = 379.477, and λ = 398.852 and λ = 408.671 nm (for lanthanum). Calibration curves were prepared using standard solutions in the concentration ranges of 0.1–1.5 mg/dm^3^ and 0.1–5 mg/dm^3^ for boron and lanthanum, respectively. Water was purified by a Millipore Elix 10 system. The changes in the chitosan structure were determined by a FTIR (Fourier-transform infrared) Spectrum Two spectrometer (Perkin Elmer, Waltham, MA, USA. SEM (scanning electron microscope) micrographs of the chitosan were collected using a Phenom ProX SEM (Phenom-World Bv, Eindhoven, The Netherlands). The phase compositions of the chitosan-based samples were determined using a Seifert 3003TT powder X-ray diffractometer with a Cu X-ray tube: kλ_1_ = 1.540598 Å, kλ_2_ = 1.544426 Å, and kβ = 1.39225 Å (Seifert, Ahrensburg, Germany). Powder samples were analyzed three times, and the XRD patterns presented in the manuscript were an average of the scans between 10° and 90° 2 Theta (step scans = 0.01°). XPS analysis was performed with PREVAC EA15 hemispherical analyzer (Prevac sp. z o.o., Rogów, Upper Silesia, Poland). Samples were irradiated with Al-Kα radiation (1486.6 eV). The pass energy was set to 200 eV for survey scans (energy step 0.8 eV) and to 100 eV for particular energy regions (energy step 0.05 eV). In order to avoid parasite charging, the charge neutralizer was applied. The spectrum was calibrated to C–C bonding at 284.8 eV [44]. The XPS peaks were resolved by curve fitting (with the use of CASA XPS software, Casa Software Ltd) with a sum of Gaussian (70%) and Lorentzian (30%) lines. For the secondary electron background the Shirley function was used. In the fitting procedure the full width at half maximum (FWHM) of the same components was allowed to vary in a narrow margin.

### 2.3. Preparation of the La-CTS Hydrogel Beads

Lanthanum-chitosan hydrogel beads were prepared according to the adopted procedure described by us previously [30]. Briefly, a chitosan solution was prepared by dissolving 3 g of chitosan in 75 cm^3^ of 1 wt% acetic acid solution. Next, 2.34 g of La(NO_3_)_3_·6H_2_O (the precursor of the hydroxide, La(OH)_3_), was added to the chitosan solution and sonicated for 30 min to obtain a well-dispersed, homogenous solution. The chitosan-precursor solution was added dropwise, using a thin needle (internal diameter of 0.9 mm), into a stirred 20 wt% aqueous NaOH solution. This resulted in the immediate coagulation and the formation of beads. The as-formed chitosan beads were then filtered and washed with deionised water to remove traces of gelling solution. The prepared chitosan attached with lanthanum was in the form of white hydrogel beads with an average diameter of about 4 mm. A portion of the chitosan beads was dried at 70 °С for 12 h to remove the water from the pore structure. The total content of lanthanum in the adsorbent was determined by the ICP-OES method after dissolving the thermally-dried chitosan beads in a concentrated HNO_3_. The dry beads were powdered in an agate mortar and sieved to a constant size < 0.45 mm. The averaged La-CTS powder was analyzed by FTIR and XRD.

### 2.4. Adsorption and Desorption Methodology

The adsorption and desorption experiments were carried out with 1 g of hydrogel beads and 0.01 or 0.02 dm^3^ of boron solution at a concentration varying in the range between 0 and 5000 mg/dm^3^, in a pH range of 4–9 and at a temperature of 20 °C for 1–72 h in a 100 cm^3^ conical flask with a ground polyurethane joint. The boron solution and hydrogel beads were shaken at a mixing rate of 60 rpm in a mechanical shaker. At the end of the experiment, the residual solution was analyzed for boron and lanthanum concentration by the ICP-OES method. Each adsorption experiment was repeated three times to obtain an average value. The content of adsorbed boron was also determined by the ICP OES method after dissolving dry adsorbent samples in concentrated HNO_3_.

The behaviour of boron and lanthanum species present together in the solution at increasing pH was also examined. For this purpose, a solution containing La(III) at a concentration of 2000 mg/dm^3^ and B(III) at a concentration of 20 mg/dm^3^ was prepared. The pH of the solution was adjusted to a value of 2 by using 0.1 M HCl. Then, 0.4 cm^3^ of the solution was used for the analysis of boron and lanthanum by the ICP-OES method. Next, 0.1 or 2 M NaOH solution was then added to increase the pH by one unit, and 0.4 cm^3^ of the solution was again taken for the analysis of the boron and lanthanum concentrations. The procedure was continued until the pH reached a value of 12.

The desorption of boron from the adsorbent was examined as follows: 1.0 g of boron-saturated hydrogel beads and 20 cm^3^ of sodium hydroxide solution, at concentrations of 0.1, 0.5, 1.0, or 1.5 mol/dm^3^, were mixed for 24 h at room temperature. The boron and lanthanum concentrations were determined by ICP-OES by taking 2.0 mL aliquots and properly diluting. The concentrations given were the means of three experimental results.

### 2.5. The Procedure of pHpzc Determination

The point of zero charge (pH_pzc_) of CTS and La-CTS hydrogel beads were determined according to Balistrieri and Murray method [45]. To a series of polyethylene bottles containing 40 cm^3^ of 0.01 M KNO_3_ solution, different volumes of either 0.1 M HCl or 0.1 M NaOH solution were added to obtain pH values from 4 to 10. The total volume of solution in each bottle was made up to 50 cm^3^ by addition of demineralized water. After 2 h of equilibration, the pH values (pH_i_) were noted and 1 g of the hydrogel beads was added to each solution. After 72 h of equilibration with discontinuous shaking, the pH value of the supernatant liquid in each bottle was noted (pH_f_). The difference between pH_i_ and pH_f_ (ΔpH) was then plotted against pH_i_. The solution pH at which ΔpH = 0 means “the pH_pzc_” of the sample.

## 3. Results

### 3.1. Calculations

The boron removal efficiency, *R* (%), and the boron adsorption capacity, *q* (mg/g), were calculated from the experimental data in each sample using the following equations:(1)$$\;R=\frac{\left(C_{0}-C\right)}{C_{0}}\times 100$$
(2)$$\;q=\frac{\left(C_{0}-C\right)}{m}\times V_{0}$$
where *C*_0_ and *C* are the initial and final concentration of boron in the solution (mg/dm^3^), respectively, *V*_0_ is the volume of adsorptive (dm^3^), and *m* is the dry mass of the adsorbent or the equivalent mass of lanthanum (g).

The percentage of boron desorption, *D* (%), was calculated from the experimental data using the following equation:(3)$$D=\frac{q_{D}}{q}\times 100$$
where *q_D_* is the amount of desorbed boron per adsorbent mass (mg/g).

To determine the margin of error, a confidence interval of 95% was calculated for each set of the samples using Microsoft Excel software.

### 3.2. La-CTS Hydrogel Beads Characteristic

#### 3.2.1. Fourier-Transform Infrared Spectrometry

Figure 1 shows the FTIR ATR spectra of unmodified chitosan (CTS) and the prepared chitosan hydrogel beads attached with La(III) particles (La-CTS). The FTIR spectrum of La-CTS shows typical bands of CTS: overlapped amide I from non-deacetylated amine groups at 1651 cm^−1^ and amide II at 1594 cm^−1^; complex saccharide ring C–H skeletal deformation at 1420 and 1379 cm^−1^; C–O–C ether group stretching at 1149 cm^−1^; secondary and primary hydroxyl or amine group stretching at 1075 cm^−1^ and 1033 cm^−1^, respectively; as well as ring breathing mode at 917 cm^−1^. Comparing the FTIR spectra of CTS and La-CTS hydrogel beads, the reduction in the intensity of the bands at 1594 cm^−1^, 1075 cm^−1^ and 1033 cm^−1^ is noticed. According to the work of Bansiwal et al. [46], this is supposedly caused by the coordination of lanthanum with –NH_2_ group. The coordination of CTS with La^3+^ shows the involvement of the chitosan amine and hydroxyl group in donor-acceptor complex formation. Furthermore, the La-CTS spectrum shows an absorption peak at 523 cm^–1^ that is attributed to the La−O stretching vibration, which exhibits that lanthanum is incorporated into CTS [41,46]. The FTIR spectrum of La-CTS-B was found to exhibit new peaks at 868 cm^−1^ and 834 cm^−1^, arising from the B–O vibration at equatorial and axial positions [47].

#### 3.2.2. Scanning Electron Microscopy

SEM images of chitosan hydrogel beads attached with La(III) particles before (La-CTS) and after (La-CTS-B) boron adsorption are shown in Figure 2. Physically, the beads look the same before and after boron adsorption. Nevertheless, SEM analysis revealed the surface morphology of the La-CTS and La-CTS-B beads to be quite different. The surface of La-CTS bead is wrinkled due to the incorporation of La(OH)_3_ into the CTS matrix. Moreover, pores and cracks are present on its surface, suggesting the high adsorption capability of this material. EDS mapping of La elemental for La-CTS bead confirms that La(OH)_3_ particles are almost uniformly embedded into the CTS matrix. After boron adsorption, however, the surface structure of the La-CTS beads clearly changes. On the surface of the La-CTS-B bead, the rippled and rough structure is replaced by a smoother and tighter structure. The results of SEM analysis indicate CTS surface structure as promising for effective boron adsorption.

#### 3.2.3. X-ray Diffractometry

Figure 3 presents the XRD patterns of the CTS hydrogel beads attached with La(III) particles before and after boron adsorption. XRD patterns of the unmodified CTS (raw material), presented in our previous work [30], indicated the amorphous character of the sample. La-CTS patterns show that the treatment of CTS with lanthanum compound leads to the formation of La(OH)_3_ crystals. The peaks at 27.40°, 27.99°, 39.47°, 48.70°, 55.27°, and several smaller peaks between 64.16° and 82.10° 2Theta correspond to lanthanum hydroxide (Reference code 00-036-1481). However, due to the presence of the broad peaks, as well as the bump between 18.90° and 24.23°, it can be concluded that the chitosan crystals are disordered. Jagtap et al. [48] have recently described the formation of biopolymer material based on chitosan flakes, also possessing lanthanum compounds. After the formation of the material, the crystals were still disordered, and only some small changes in the shape of the XRD pattern were recorded. The materials were characterized using the FTIR spectra, which confirmed the formation of hydroxyl groups onto the chitosan, which were probably related to lanthanum ions. In our case, the formation of the lanthanum hydroxide phase was confirmed using the XRD technique.

The additional crystals were formed after boron adsorption (Figure 3). Apart from the lanthanum hydroxide, the hydrogen borate was also found. The small peaks at 18.05°, 20.21°, 22.37°, 38.70°, 48.92°, 80.70°, 54.61° 2Theta indicate the formation of HBO_2_ powder (reference code 00-015-0403). A similar result of the HBO_2_ formation and analysis of the phase composition of material used for the removal of fluoride from aqueous solution was presented by Ranjan et al. [49]. In that case, the materials were not blended with the chitosan, and the XRD patterns indicated that the crystals were in much better order.

#### 3.2.4. Inductively Coupled Plasma Optical Emission Spectrometry

The content of lanthanum in La-CTS hydrogel beads, determined by ICP OES analysis after dissolving three dry samples in concentrated HNO_3_, was equal to 1.36 ± 0.05%. This confirmed once again the presence of lanthanum in the La-CTS hydrogel beads.

#### 3.2.5. X-ray Photoelectron Spectroscopy

Figure 4 presents the high-resolution scans of the XPS regions of interest; the rest of the energy regions are not shown for brevity. The particular components were denoted with Arabic numbers. Panels (a) and (b) present decomposed La 3d5/2 regions. In the case of the La-CTS sample (panel (a)), only two components are present in the spectra; the most likely assignment is (1) La-NH_2_ and (2) La–OH/La–OH(H_2_O). In the case of the La-CTS-B sample (panel (b)), the third distinguishable component appears, and most likely can be assigned as (3) La-O-B configuration. This is most likely due to fact that no other element was introduced into the CTS sample. Panel (c) presents the B 1s energy region recorded for La-CTS-B sample. The spectrum confirms the introduction of B into the structure, however the significant shift towards higher binding energies can be observed, as for highly oxidized boron species [50]. Basing on the available databases, the component assignment we propose here is (4) La–O–B; (5) B–OH (6) is the plasmon energy loss feature. The component energy positioning is consistent with available databases [51].

#### 3.2.6. Point of Zero Charge Determination

A point of zero charge on a surface (pH_pzc_) means that the surface of the sorbent has a positive charge below pH_pzc_ and negative charge above pH_pzc_. The determined pH_pzc_ value of CTS is equal to 7.4 ± 0.1, whereas for La-CTS this value is 8.8 ± 0.1 (Figure 5). In the case of the adsorptive removal of anions from the aqueous solution, it is desirable for the sorbent surface to be positively charged. Accordingly, at pH < 8.8 (a value of pH_pzc_), the surface of La-CTS is positively charged, and B(OH)_4_^−^ species, which are negatively charged, can be electrostatic attracted hydrogel surface.

### 3.3. The Influence of pH on Boron Adsorption

The dominant forms of inorganic boron in aqueous systems are mononuclear species, such as boric acid H_3_BO_3_ (or B(OH)_3_) and oxyborate ion H_2_BO_3_^−^ (or B(OH)_4_^−^). The distribution of these forms is connected with the first dissociation constant K_a_ of boric acid that is equal to 5.8 × 10^−10^ mol/dm^3^ in pure water at a temperature of 25 °C [1]. The most important parameter, which determines the speciation of boric acid and borate ion in aqueous solutions, is the pH of the medium. As it can be seen in Figure 6, the boric acid molecule dominates at low and neutral pH values, whereas the borate monovalent anion dominates at high pH values [52].

The usual lanthanum oxidation state is +3. In an aqueous solution, lanthanum is present as hydrated ions, [La(H_2_O)*_n_*]^3+^ (*n* > 6), which hydrolyse according to the following reaction [53]:[La(H_2_O)*_n_*]^3+^ + (*n* − *m*) H_2_O → [La(OH)*_m_*(H_2_O)*_n_*_−*m*_]*^m^*^−3^ + (*n* − *m*) H_3_O^+^(4)
where *m* is 1, 2, or 3, and finally may form lanthanum(III) hydroxide (when *m* = 3) according to the following equation:[La(OH)_3_(H_2_O)*_n_*_−3_] → La(OH)_3↓_ + (*n* − 3) H_2_O(5)

At pH above 6, La(OH)_3_ is a white gelatinous precipitate that readily solubilizes in acids, while it becomes insoluble in excess bases [54]. Its adsorption properties towards boron are shown in Figure 7 (Graph 1). Because of the developed surface of freshly precipitated lanthanum hydroxide, boron adsorption is rather good. As can be seen in Figure 7, the pH value is a significant parameter affecting boron removal. The process of boron adsorption started at pH 7 to reach a maximum value of *R* = 58 ± 1.1% (corresponding to *q* = 5.8 mg/g_(La)_) at pH 9.8. The adsorption was observed to decrease with a further increase in pH. Graph 2 in Figure 7 shows how the adsorption of boron on chitosan modified with lanthanum(III) changed with the pH of the solution, reaching the maximum boron adsorption of 6.14 ± 0.41 mg/g (calculated per 1 g of La) at pH 5.0 when the neutral monomolecule, B(OH)_3_, dominates in the solution (Figure 6). At pH < 5, the adsorption of boron on La-CTS was slightly lower than at pH 5.0, but when the pH increased from 5 to 9, boron removal on La-CTS beads rapidly decreased. Importantly, no elution of La(OH)_3_ from hydrogel was observed at acidic pHs when an initial concentration of boron was 20 mg/dm^3^.

### 3.4. Kinetic Studies

The effect of contact time, *t*, on boron adsorption is shown in Figure 8. As can be observed, the equilibrium state was reached after 24 h, independently of the initial boron concentration. Considering the adsorption process from the solid phase (*q_t_* = *f*(*t*)) it can be described by four kinetic first-order and second-order models, both in reversible and irreversible forms [55]. In this study, two forms of irreversible kinetics, namely the pseudo-first-order (PFO) and the pseudo-second-order (PSO) equations, commonly used to describe adsorption data, were tested [56].

The pseudo-first-order kinetic equation is represented by Equation (6):(6)$$\frac{dq_{e}}{dt}=\;k_{1}\left(q_{e}-q_{t}\right)$$

The pseudo-second-order (PSO) kinetic equation is represented by Equation (7):(7)$$\frac{dq_{e}}{dt}=\;k_{2}{\left(q_{e}-q_{t}\right)}^{2}$$
where *q_t_* and *q_e_* are the amounts of boron adsorbed (mg/g) at any time *t* and at the time of equilibrium, while *k*_1_, and *k*_2_ are the pseudo-first-order and pseudo-second-order rate constants, respectively. Modeling calculations were conducted using RStudio software by means of a non-linear regression method based on the Levenberg–Marquardt algorithm.

The resulting parameters along with the standard deviation and the correlation factor *R*^2^ of the PFO and PSO kinetic models for various initial boron concentrations are listed in Table 1. The calculated value of *R*^2^ for the pseudo-first-order equation was lower than for the pseudo-second-order equation, indicating that the PFO model was not applicable to the investigated process. For the PSO model, the coefficient *R*^2^ was in the range between 0.955 and 0.995. The calculated values of *q_e_* agreed with the experimental adsorption capacities, *q_expt_*, when the initial boron concentrations were 20 and 200 mg/dm^3^. Thus, the constant *k*_2_ could be used to calculate the initial adsorption rate, *r*, at *t* = 0, defined by Equation (8):(8)$$\;r=\;k_{2}\times {\left(q_{e}\right)}^{2}$$

The initial adsorption rate, *r*, increased when the initial boron concentration increased (Table 1). The results indicated that the PSO model was appropriate to describe boron adsorption on the solid surfaces of the La-CTS hydrogel, except for very high concentrations of boron. Therefore, we expect that the adsorption process was probably heterogeneous, mainly because of transport phenomena and the chemical reactions’ inseparability.

### 3.5. Isotherm Models

To assess the maximum adsorption capacity and to determine the mechanism of adsorption, two-parameter models of isotherms describing experimental data are used: the Langmuir, Freundlich, Dubinin–Radushkevich, and Temkin equations [57,58]. The most developed equations used to describe adsorption equilibria are mainly the Langmuir and Freundlich. The Langmuir model assumes that adsorption occurs on surface sites on each, of which the energy is equal, while the Freundlich model allows for several kinds of adsorption sites in the solid, each having different energy of adsorption. The Dubinin–Radushkevich isotherm is a local isotherm also often used to describe adsorption, explaining the equilibrium of adsorption by the theory of volume filling of adsorbent micropores. The Temkin isotherm describes monolayer adsorption on a heterogeneous surface, assuming that the heat of adsorption of all molecules in the layer decreases linearly due to the adsorbent-adsorbate interaction, and the adsorption is characterized by an even distribution of binding energies.

The Langmuir equation is represented by Equation (9):(9)$$q_{e}=q_{m}\times \frac{B\times C_{e}}{1+B\times C_{e}}$$

The Freundlich equation is represented by Equation (10):(10)$$q_{e}=K_{F}\times {\left(C_{e}\right)}^{1/n}$$

The Dubinin–Radushkevich equation is represented by Equation (11):(11)$$q_{e}=\left(X_{m}\right)\;exp\left(-k\times \varepsilon^{2}\right)$$

The Temkin equation is represented by Equation (12):(12)$$q_{e}=\frac{R\times T}{b_{T}}ln\;\left(A_{T}\times \;C_{e}\right)$$
where *C_e_* is the concentration of boron in the solution in equilibrium (mg/dm^3^); *q_m_* (mg/g) and *B* (dm^3^/mg) are the Langmuir parameters; *K_F_* (mg/g) and *n* are the parameters resulting from the Freundlich model; *ε*, Polanyi potential, *X_m_* (mol/g) and *k* (mol^2^/kJ^2^) are the Dubinin-Radushkevich parameters; the parameters b_T_ (kJ/mol) and A_T_ (dm^3^/mg) are calculated from the Temkin model; *R* is the gas constant (8.314 J/(mol∙K)); *T* is the temperature (K). Non-linear regression method (the Levenberg–Marquardt algorithm) was used to determine the best fit of equilibrium data. The isotherm parameters obtained, along with the standard deviation (SD) and correlation factor (*R*^2^), are listed in Table 2.

Figure 9 shows the adsorption isotherm of boron on the La-CTS hydrogel beads and the fitting of the Langmuir, Freundlich, Dubinin–Radushkevich, and Temkin adsorption models onto the experimental data. The experimental adsorption capacity, *q*, was estimated as 11.1 ± 0.3 mg/g (calculated per 1 g of dry La-CTS-B) at 20 °C, with a contact time of 24 h and an initial boron concentration of 100 mg/dm^3^. Furthermore, the content of boron in the thermally dried La-CTS-B (after dissolving three samples in concentrated HNO_3_) was equal to 11.0 ± 0.2 mg/g, which confirmed the experimental adsorption capacity calculated according to Equation (1).

The obtained value of adsorption capacity is a high result compared to value *q* equal 3.3 mg/g for unmodified chitosan hydrogel [59] and for chitosan-based adsorbents filled with various additives. For example, chitosan hydrogels modified with the metal oxide nanoparticles, CrO_3_, TiO_2_, and Fe_3_O_4_, showed sorption capacities of 3.5, 4.3, and 4.4 mg/g, respectively, and chitosan hydrogels modified with cobalt, nickel, and iron hydroxides showed sorption capacities of 2.5, 61.4 (monolayer capacity), and 7.8 mg/g, respectively [27,30,33]. Gazi and Shahmohammadi noted very good results of 23.8 mg/g using glycidol modified chitosan, whereas Wei et al. reported 35.1 mg/g sorption capacity on *N*-methyl-d-glucamine functionalized chitosan [60,61]. Furthermore, it was found that the La-CTS was a better adsorbent than the gelatinous precipitate of lanthanum hydroxide. The latter had a boron adsorption capacity equal 5.80 ± 0.11 mg/g_(La)_, while for La-CTS this parameter was equal to 6.14 ± 0.41 mg/g_(La)_, starting from the same initial boron concentration (20 mg/dm^3^) as described in Kinetic Studies section. Additionally, the fine and difficult-to-separate precipitate of lanthanum hydroxide could be disadvantageous in water treatment technology, in contrast to La-CTS in the form of the hydrogel beads proposed in this study.

The results of the boron adsorption modelling, especially such parameters as Langmuir parameters (*q_m_* and *B*), adsorptive capacity (*q_m_*, mg/g), expressed as the maximum amount of boron that can be adsorbed by the adsorbent as a monolayer, as well as equilibrium constant that corresponds to the adsorption energy (*B*, dm^3^/mg), are presented in Table 2. The parameters *K_F_* (mg/g) and *n* resulting from the Freundlich model correspond to the relative adsorptive capacity and the adsorption intensity of the adsorbent, respectively. The essential characteristics of a Langmuir isotherm can be expressed in terms of a dimensionless constant separation factor, *R**_L_*, which is defined by Equation (13):(13)$$R_{L}=\frac{1}{1+B\times C}$$

According to Hall et al. [62], the parameter *R**_L_* indicates the shape of the isotherm in the following manner: *R**_L_* > 1, unfavourable; *R**_L_* = 1, linear; 0 < *R**_L_* < 1, favourable; and *R**_L_* = 0, irreversible. Similarly, the goodness of the fit using the F equation to describe the adsorption can be assessed by the constant *n*. If 1 < *n* < 10, the F equation is adequate to use [63].

In the Dubinin–Radushkevich equation, the *X_m_* parameter is the adsorption capacity (mol/g) and the *k* parameter is a constant related to the adsorption energy (mol^2^/kJ^2^). The adsorption energy, *E* (the energy required to transfer 1 mol of adsorbate species to the surface of the adsorbent from infinity in the bulk of the solution) is obtained from the following Equation (14):(14)$$\;E=-{\left(2\times k\right)}^{-0.5}$$

If the energy of adsorption is less than 20 kJ/mol, the adsorption is physical in nature due to weak van der Waals forces. The energy for chemisorption lies in the range 40–800 kJ/mol [63]. The parameters *b_T_* (kJ/mol) and *A_T_* (dm^3^/mg), calculated from the Temkin model, correspond to the adsorption energy change and the adsorption equilibrium constant, respectively.

As it can be seen in Table 2, the Freundlich model gave the best fit to the experimental data of boron adsorption on La-CTS hydrogel because of its highest correlation coefficient (0.999). The *K_F_* parameter was equal to 0.508 mg/g, while the parameter *n* was equal to 1.35, indicating the adequacy of the Freundlich model for describing the investigated process.

A slightly lower correlation coefficient (0.997) was obtained by fitting the Langmuir model. The dimensionless constant separation factor, *R**_L_*, in the range between 0.476 and 0.978 for initial boron concentrations of 2–100 mg/dm^3^, indicated favourable adsorption. Unfortunately, the value of parameter *q_m_*, different from the experimental capacity, *q_expt_*, indicated the impossibility of the full interpretation of the process as monolayer adsorption.

A slightly less-adequate fit was obtained for the Dubinin-Radushkevich model (correlation coefficient of 0.924). The calculated value *X_m_* (10.2 ± 1.5 mg/g) was in a good agreement with the experimental *q_expt_*, 11.1 ± 0.3 mg/g. The other significant parameter was adsorption energy determined on the basis of the Dubinin-Radushkevich model, which was calculated as 0.12 kJ/mol, indicating a physical process in nature. The worst fit was obtained for the Temkin model, which was inadequate for describing boron adsorption on La-CTS.

Generally, the Freundlich equation fits the experimental data better than the other models because of its high correlation coefficients. The Freundlich model usually corresponds to the adsorption process on heterogeneous surfaces. The better conformity of the Freundlich model over the other isotherm models for boron adsorption was noted by Bursali et al. [59] on freeze-dried chitosan beads, by Wei et al. [61] using freeze-dried chitosan functionalised with N-methylglucamine, and by Kluczka et al. [30] on cobalt(II) doped chitosan hydrogel.

### 3.6. Mechanism of La-CTS Formation and Boron Adsorption

In this study, CTS flakes and lanthanum(III) nitrate were used as precursors of the La-CTS hydrogel beads. When the pH of the solution is lower than the dissociation constant of CTS (pK_a_ = 6.3–6.8), which occurs during the dissolving of CTS in 1% acetic acid, the amine groups are protonated. After the addition of La(NO_3_)_3_ to CTS solution, it solubilizes and undergoes hydration. Under the influence of the NaOH solution, the gelling process starts and the lanthanum complex can coordinate with the reactive amine groups of CTS [41,43]. This results in the formation of a lanthanum-amine adduct and a modification of CTS structure (Scheme 1).

According to our results, the most beneficial adsorption of boron on freshly precipitated lanthanum(III) hydroxide, La(OH)_3_, was noted at pH 9.8 (Graph 1 in Figure 7) when the majority of borate anions, [B(OH)_4_^−^], is observed (Figure 6). Therefore, it can be stated that the process occurred via chemical reaction following electrostatic attraction between the positively charged surface of the hydroxide and the negative boron ions, according to Equation (15):(15)$${\left(H_{2}O\right)}_{n-2}-{\mathrm{La}}^{⨁}\langle \begin{array}{c} \mathrm{OH}\\ \mathrm{OH} \end{array}+\begin{array}{c} \mathrm{HO}\\ \mathrm{HO} \end{array}\rangle B^{⊖}<\begin{array}{c} \mathrm{OH}\\ \mathrm{OH} \end{array}↔{\left(H_{2}O\right)}_{n-2}\mathrm{HO}-\mathrm{La}\langle \begin{array}{c} O\\ O \end{array}\;\rangle B-\mathrm{OH}+2H_{2}O\;\;\;\;\;\;$$

Nevertheless, the mechanism of boron adsorption on La-CTS was different than for La(OH)_3_. Knowing that the maximum boron adsorption on hydrogel was observed at pH of 5 when the majority of boric acid in the solution is present as a neutral monomolecule, B(OH)_3_ (Figure 6), and taking into account suggestions resulting from FTIR, XRD, and XPS analyses, it could be assumed that the dominant mechanism of adsorption was the interaction between the boric acid molecule and hydroxyl groups bound to the lanthanum in complex with CTS (Scheme 2).

### 3.7. The Stability of the La-CTS and Desorption Tests

To qualify La-CTS as a suitable bio-adsorbent for drinking water treatment, the stability of hydrogel beads, and in particular the bond strength between lanthanum and CTS, was considered. For this purpose, the elution of lanthanum from the La-CTS was tested under the conditions of the adsorption and desorption processes.

The effect of contact time and initial boron concentration on the elution of lanthanum ions from the La-CTS hydrogel beads, as well as on the content of lanthanum in La-CTS during the adsorption at pH 5, is presented in Figure 10. The graph 1 shows that La(III) ions did not elute from the hydrogel, even after 3 days of contact time of La-CTS and adsorptive, when the initial concentration of boron was less than 500 mg/dm^3^. The elution of the lanthanum(III) in the concentration range from 9.24 ± 0.18 mg/dm^3^ to 19.2 ± 0.3 mg/dm^3^ was noticeable when the initial boron concentration was equal to, or exceeded, 500 mg/dm^3^, and when the contact time was elongated to 3 days. As it can be seen in Figure 10, the increase in the lanthanum concentration in the eluate was accompanied by a reduction in the lanthanum content in the La-CTS. This can be easily observed when comparing bars in graph 2, which varied from 1.20 ± 0.06% to 0.99 ± 0.06%, while the boron concentration increased from 500 to 5000 mg/dm^3^, and the contact time was 72 h. Because the optimal adsorption time was set to 24 h (see Figure 8), and after this time no leaching of the lanthanum from the hydrogel was found, it can be concluded that the proposed bio-adsorbent was stable under the conditions of the adsorption process.

Next, the possibility of boron desorption from the La-CTS hydrogel beads without the loss of lanthanum in the adsorbent was considered. Because the adsorption capacity of boron(III) on La-CTS was relatively low under alkaline conditions (see Figure 7) and the lanthanum hydroxide is insoluble in excess base, boron desorption process was carried out using various concentrations of sodium hydroxide solution. The effect of the NaOH concentration on boron desorption and on lanthanum elution is presented in Figure 11. As it can be seen, satisfactory boron desorption was obtained with 1 M NaOH solution. Preferably, only 0.012 ± 0.009 mg/dm^3^ La was found in the regenerative solution after desorption. The obtained results indicate that the La-CTS hydrogel beads can be effectively regenerated and re-used in the next adsorption cycles.

## 4. Conclusions

Lanthanum(III)-attached chitosan beads (La-CTS) have been prepared by the method employing co-precipitation and coagulation for the elimination of the excess of boric acid from modelling solutions. La-CTS was shown to possess higher adsorption capacity than lanthanum hydroxide, La(OH)_3_, and the precursors of the proposed bio-adsorbent, namely native chitosan hydrogel beads. Such high efficiency of the chitosan-La(III) complex for boron(III) recovery was related to the fact that hydroxyl groups, bound to the lanthanum centre in this complex, can interact with HBO_2_ species.

La–CTS was stable under the conditions of the adsorption-desorption process. In the near future, the dynamic capacity of La-CTS in the flow system will be examined, as well as its selectivity in removing boron from the real media, e.g., high-salinity solutions. We strongly believe that the proposed La-CTS hydrogel beads could serve as an advantageous alternative for the effective separation of boron(III) from polluted water.

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
