# Peer review of "The Use of Lanthanum Ions and Chitosan for Boron Elimination from Aqueous Solutions"

_polymers, 2019, doi:10.3390/polym11040718_

Round 1
Reviewer 1 Report
The paper presents a routine adsorption study in which aqueous boron was removed using lanthanum-modified chitosan. As authors state in the introduction, the use of lanthanum for boron removal has not been studied. From practical point of view, however, the price of lanthanum-materials is prohibitive in many cases for real water treatment uses. Nevertheless, the work is generally conducted well and English grammar and style are fine.
Some comments and questions needs to be though clarified before considering acceptance:
1. Authors should mention in the introduction one commercial adsorbent-product based on lanthanum: Phoslock (many articles available if you search for instance Scopus). It is bentonite clay modified with lanthanum and intended for phosphorus removal.
2. Figure 2: It is not necessary to show the spectrum, so C-part of the figure can be removed.
3. Figure 3: This should be edited to contain both of the diffractograms in one figure.
4. Line 217: Lanthanum in your study is attached to chitosan. Why do you discuss here dissolved lanthanum? Or is it leaching out of the chitosan?
5. Figure 4 is unclear to read. What part of the figure shows removal efficiency?
6. Authors should add speciation diagram of boron at different pH. This would help to understand the observed trends.
7. Use nonlinear kinetics equations (i.e., nonlinear regression) to obtain the numerical values of parameters. The use of linear equations is considered widely obsolete as it introduces errors.
8. Line 290: Higher than adsorption of what? You state in the introduction that this kind of adsorbent has not used earlier for boron removal. Add also reference here.
9. Figure 6: Replace "Model L" with Langmuir, "Model F" with Freundlich etc.
10. Similarly as with kinetics, the use of linearized form of adsorption isotherms is considered obsolete. Use nonlinear forms and nonlinear regression.
11. Figure 7: It is unclear which part of the figure refer to the left axis and which part to the right axis. Also: what is "1.36 +/- 0.05%"?
12. Did adsorbent remain active after the desorption tests shown in Figure 8?
13. Lanthanum is relatively expensive. Please add some considerations about the costs as well.
Author Response
Manuscript ID: polymers-474524
Title: The use of lanthanum ions and chitosan for boron elimination from aqueous solutions
Authors: Joanna Kluczka*, Gabriela Dudek, Alicja Kazek-Kęsik, Małgorzata Gnus, Maciej Krzywiecki, Krzysztof Mitko and Katarzyna Krukiewicz
RESPONSE TO REVIEWER #1
Thank you for your comments and suggestions that allowed us to greatly improve the quality of the manuscript.
1. Authors should mention in the introduction one commercial adsorbent-product based on lanthanum: Phoslock (many articles available if you search for instance Scopus). It is bentonite clay modified with lanthanum and intended for phosphorus removal.
Answer: Thank you for this comment. We have added some information about Phoslock to the introduction and completed adequate references in the revised manuscript.
The revised manuscript has been improved as follows:
Lanthanum is used to produce Phoslock, the commercial bentonite clay in which the sodium and/or calcium ions are exchanged for La3+. The main application of Phoslock is the reduction of phosphorus, one of the major contributing factors to algal growth in lakes, ponds and drinking water storage reservoirs [Spears, B.M.; Lürling, M.; Yasseri, S.; Castro-Castellon, A.T.; Gibbs, M.; Meis, S.; McDonald, C.; McIntosh, J.; Sleep, D.; Van Oosterhout, F. Lake responses following lanthanum-modified bentonite clay (Phoslock®) application: An analysis of water column lanthanum data from 16 case study lakes. Water Res., 2013 47, 5930-5942, https://doi.org/10.1016/j.watres.2013.07.016.; Wang, C.H.; Wu, Y.; Wang, Y.Q.; Bai, L.L.; Jiang, H.L.; Yu. J.H. Lanthanum-modified drinking water treatment residue for initial rapid and long-term equilibrium phosphorus immobilization to control eutrophication. Water Res., 2018 137, 173-183.].
2. Figure 2: It is not necessary to show the spectrum, so C-part of the figure can be removed.
Answer: We have removed this figure.
3. Figure 3: This should be edited to contain both of the diffractograms in one figure.
Answer: We have done it. The diffraction patterns are in one figure in the revised manuscript.
4. Line 217: Lanthanum in your study is attached to chitosan. Why do you discuss here dissolved lanthanum? Or is it leaching out of the chitosan?
Answer: Indeed, it may be surprising that we have included the study on the co-precipitation/adsorption of borates on the "in situ" hydroxide in our manuscript. However, we needed this result to compare with sorption on La-CTS. Unfortunately, we did not find enough data in other articles.
Lanthanum is not leaching out of the La-CTS. Neither during adsorption (Figure 10) nor during desorption tests (Figure 11).
5. Figure 4 is unclear to read. What part of the figure shows removal efficiency?
Answer: We agree with the reviewer. This figure is not clear enough. We have improved it in the revised manuscript. 1- “Boron removal efficiency on La(OH)3” whereas 2 – “Boron adsorption capacity on La-CTS” (Figure 7 in the revised manuscript).
6. Authors should add speciation diagram of boron at different pH. This would help to understand the observed trends.
Answer: Thank you for this comment. We have included Figure 6 shown boron speciation at different pH.
7. Use nonlinear kinetics equations (i.e., nonlinear regression) to obtain the numerical values of parameters. The use of linear equations is considered widely obsolete as it introduces errors.
Answer: Thank you for your advice. We used a nonlinear least squares regression (the Levenberg–Marquardt algorithm) to identify the best kinetic model. The same in the case of isotherm equations.
8. Line 290: Higher than adsorption of what? You state in the introduction that this kind of adsorbent has not used earlier for boron removal. Add also reference here.
Answer: In the revised manuscript we have rebuilt this discussion as follows:
The obtained value of adsorption capacity is a high result compared to value q equal 3.3 mg/g for unmodified chitosan hydrogel [59] and for chitosan based adsorbents filled with various additives. For example, chitosan hydrogels modified with the metal oxide nanoparticles, CrO3, TiO2 and Fe3O4, showed sorption capacity 3.5, 4.3 and 4.4 mg/g, respectively, and chitosan hydrogels modified with cobalt, nickel and iron hydroxides showed sorption capacity 2.5, 61.4 (monolayer capacity) and 7.8 mg/g, respectively [27,30,33]. Gazi and Shahmohammadi noted very good results 23.8 mg/g using glycidol modified chitosan whereas Wei et al. reported 35.1 mg/g sorption capacity on N-methyl-D-glucamine functionalized chitosan [60,61].
9. Figure 6: Replace "Model L" with Langmuir, "Model F" with Freundlich etc.
Answer: We have replaced the labels of models to the full name.
10. Similarly as with kinetics, the use of linearized form of adsorption isotherms is considered obsolete. Use nonlinear forms and nonlinear regression.
Answer: Thank you once again for this comment. We have done it.
11. Figure 7: It is unclear which part of the figure refer to the left axis and which part to the right axis. Also: what is "1.36 +/- 0.05%"?
Answer: We agree with the reviewer that this figure is not clear. We have tried to improve it in the revised manuscript.
"1.36 +/- 0.05%" is a content of lanthanum in La-CTS beads before adsorption. We removed this value from the figure.
12. Did adsorbent remain active after the desorption tests shown in Figure 8?
Answer: Yes, its ability to perform the next sorption cycle has been checked twice. However, we did not carry out more tests. We plan to do it in a column study.
13. Lanthanum is relatively expensive. Please add some considerations about the costs as well.
Answer: We agree with the reviewer that lanthanum is not a cheap element. However, its content in the proposed sorbent is small, and chitosan is a cheap material. Furthermore, the encapsulation technique of the hydroxide-CTS hybrids preparation is reported to be a low-cost method [26,27]. Hence, the cost of La-CTS would not be higher than the mentioned Phoslock. And Phoslock is used on a large scale in lake restoration projects.

Reviewer 2 Report
the manuscript can be accept to publish after a minor revision.
1. page 6, line 204,,, “---were formed after boron adsorption (Figure 1B). ” there was not figure 1 B in the manuscript.
2. Table 1 was not properly presented, please change its format.
Author Response
Manuscript ID: polymers-474524
Title: The use of lanthanum ions and chitosan for boron elimination from aqueous solutions
Authors: Joanna Kluczka*, Gabriela Dudek, Alicja Kazek-Kęsik, Małgorzata Gnus, Maciej Krzywiecki, Krzysztof Mitko and Katarzyna Krukiewicz
RESPONSE TO REVIEWER #2
Thank you very much for your comments and suggestions that allowed us to improve the quality of the manuscript.
1. page 6, line 204,,, “---were formed after boron adsorption (Figure 1B). ” there was not figure 1 B in the manuscript.
Answer: Thank you for this comment. Of course there was not figure 1B. It should be figure 3. In the revised manuscript we have improved this number. Furthermore the diffraction patterns are in one figure in the revised manuscript as suggested the reviewer no. 1.
2. Table 1 was not properly presented, please change its format.
Answer: We agree with the reviewer. This table is not clear enough. It has been removed in the revised manuscript. The equations have been placed in the text.

Reviewer 3 Report
Comments:
1) Authors wrote: “Chitosan (CTS) is a natural biopolymer derived from the alkaline deacetylation of chitin” – in this sentence you already deny that chitosan is natural biopolymer. Please note that chitosan is not a natural biopolymer – there are no organisms which consists chitosan. All known organisms consist chitin – chitin is natural biopolymer. Chitosan is therefore a derivative of chitin that can be obtained via chemical or biological deacetylation. Probably you can rewrite the sentence that chitosan is a biorenewable polymer.
2) Describe how chitosan can bind various metals, metalloids and how can it be used in modern technologies.
Examples: 10.1007/s10570-016-0962-6; 10.1016/j.snb.2018.05.173
3) To support the proposed mechanism of interactions between chitosan, lanthanium and boric acid please provide the XPS analysis of chitosan; chitosan-La; chitosan-La-boron
4) Additionally, for better understanding it would be beneficial to analyze the zeta potential of chitosan and chitosan-La.
Author Response
Manuscript ID: polymers-474524
Title: The use of lanthanum ions and chitosan for boron elimination from aqueous solutions
Authors: Joanna Kluczka*, Gabriela Dudek, Alicja Kazek-Kęsik, Małgorzata Gnus, Maciej Krzywiecki, Krzysztof Mitko and Katarzyna Krukiewicz
RESPONSE TO REVIEWER #3
Thank you for your comments and suggestions that allowed us to greatly improve the quality of the manuscript.
1) Authors wrote: “Chitosan (CTS) is a natural biopolymer derived from the alkaline deacetylation of chitin” – in this sentence you already deny that chitosan is natural biopolymer. Please note that chitosan is not a natural biopolymer – there are no organisms which consists chitosan. All known organisms consist chitin – chitin is natural biopolymer. Chitosan is therefore a derivative of chitin that can be obtained via chemical or biological deacetylation. Probably you can rewrite the sentence that chitosan is a biorenewable polymer.
Answer: Thank you very well for this comment. Of course, you are right. I have changed the sentence as you suggested.
2) Describe how chitosan can bind various metals, metalloids and how can it be used in modern technologies.
Examples: 10.1007/s10570-016-0962-6; 10.1016/j.snb.2018.05.173
Answer: Thank you for this comment. We have written more about chitosan and its bindings with metal and metalloid ions in the introduction section. We have shortly described its applications. Finally, we have completed adequate references in the revised manuscript according to your suggestion.
3) To support the proposed mechanism of interactions between chitosan, lanthanium and boric acid please provide the XPS analysis of chitosan; chitosan-La; chitosan-La-boron
Answer: Thank you for your advice. As you suggested we have performed the XPS analysis – see Figure 4. The description you find in 2.2 section entitled “Analysis and Apparatus” and 3.2.5 section entitled “X-ray photoelectron spectroscopy”. As a result of this analysis we confirmed the formation of La-CTS hydrogel beads (Scheme I) and proposed mechanism of adsorption shown in scheme II. Thank you once again.
4) Additionally, for better understanding it would be beneficial to analyze the zeta potential of chitosan and chitosan-La.
Answer: Thank you for your comment. The point of zero charge (pHpzc) of a CTS and La-CTS hydrogel beads was determined according to Balistrieri and Murray method [44]. This procedure has been described in 2.5 section while results have been presented in Figure 5 and 3.2.6 section.

Round 2
Reviewer 1 Report
After these edits, the manuscript can be accepted.
Reviewer 3 Report
Accept